# Uvula infections and traditional uvulectomy: Beliefs and practices in Luwero district, central Uganda

**Simon Peter Sebina Kibira**[1]*, **Juliana Namutundu**[2], **Julius Kiwanuka**[2], **Noah Kiwanuka**[2], **Victoria Nankabirwa**[2], **Justine Namwagala**[3]

1 Department of Community Health and Behavioural Sciences, School of Public Health, College of Health Sciences, Makerere University, Kampala, Uganda, 2 Department of Epidemiology and Biostatistics, School of Public Health, College of Health Sciences, Makerere University, Kampala, Uganda, 3 Department of Ear Nose and Throat, School of Medicine, College of Health Sciences, Makerere University, Kampala, Uganda

* pskibira@musph.ac.ug

**Data Availability Statement:** Excerpts of the transcripts related to this paper are available as supporting information file.

## Abstract

Uvulitis is the inflammation and swelling of the uvula, usually associated with infection of nearby structures. Uvulitis can be treated symptomatically, using medication or in some cases with uvulectomy, the uvula surgical removal or shortening. Traditional uvulectomy by traditional practitioners has been practiced in Africa for ages, associated with adverse outcomes. Although there is no empirical evidence for the association between adverse outcomes and traditional uvulectomy in Uganda, anecdotal findings showed incidents of uvula infections following uvulectomy in central Uganda. While these findings also indicate that traditional uvulectomy is common, the community understanding of uvulitis, the beliefs and practices are not well understood. This qualitative study sought to understand beliefs and practices using interviews with community health workers, traditional uvulectomy clients, and traditional surgeons, and focus group discussions with community members. Transcribed data were analysed in Atlas.ti 9 using thematic analysis steps. The findings show that uvula infection, locally known as "*Akamiro*" and the associated traditional uvulectomy are common in Luwero and beyond. "*Akamiro*" was described as larger than the normal, the size of a chicken heart or a big pimple, visible when a child cries, with unknown causes. Symptoms included persistent cough, diarrhoea, vomiting, loss of appetite, inability to swallow and ultimately weight loss, swollen stomach, saliva overflow, fever, breathing and speech difficulty. Diagnosis was confirmed after seeking care from health workers or in consultation with significant others and finally the traditional surgeon; in a hierarchical pattern. Uvulectomy was conducted by traditional surgeons, with surgery lasting a few minutes, in the morning or after sun-set. Tools used were razor blades, reeds, strings, wires, sickle knives and spoons. Payment was flexible; cash or in-kind. Surgeons had immense community trust, including community health workers. Interventions to support persons with uvula infections need to address the health system weaknesses, and health education.

**Funding:** This work was supported by the Government of Uganda through the Makerere University Research and Innovations Fund to JN and NK. The funder had no role in the design, data collection or interpretation of the findings.

**Competing interests:** The authors have declared that no competing interests exist.

## Introduction

The uvula is a small piece of soft tissue that dangles down centrally from the soft palate. It has an abundance of salivary glands, and plays a large role in moistening the oropharyngeal and pharyngeal mucosa. The Uvula is also important for articulation of speech [1]. This tissue can sometimes be inflamed and swollen leading to unpleasant gagging or chocking at the base of the tongue, a condition called Uvulitis [2, 3]. Uvulitis has been attributed to infectious and non-infectious causes. The common infectious causes are Streptococcus Group A associated with pharyngitis and tonsillitis [4] and *Hemophilus Influenzae* type B (Hib) [5–7]. In rare cases, uvulitis may be caused by *Streptococcus Pneumoniae* [8] and *Candida Albicans* [9]. The non-infectious causes of uvulitis include; trauma, chemical irritation, inhalation of cannabinoids due to use of marijuana, angioedema of either immunologic or nonimmunologic mechanisms, allergy, dehydration, snoring, drugs, inhalational irritants, neoplasia [10] and in very rare cases the sting of an inhaled bee [11].

Mild cases of uvulitis can be treated symptomatically especially in absence of fever and airway compromise [12]. In presence of infection, medications such as antibiotics, corticosteroids, diphenhydramine, and B-agonists can be used to treat uvulitis [13]. In some cases, uvulitis has been treated by uvulectomy, the surgical excision of the uvula [14–17]. Traditional uvulectomy; the removal of part or whole of the uvula by traditional practitioners using non-medical means, is one of the reported remedies for uvulitis. In some cases, it has been practiced as a ritual even without presence of symptoms [18]. This is an age-old practice that has been reported in earlier [18, 19] and more recent studies in Africa [20–24] including the burden of its adverse consequences, among both adults and children [25]. Some studies have recommended eliminating this as a dangerous practice [24–27] but it continues to be carried [24] in several areas.

In Uganda, until the current study, to the best of our knowledge, there was no published report of traditional uvulectomy. An informal investigation conducted in 2019 by some of the co-authors NK, VN, and JN, revealed the practice could have been widespread. This followed an alert by a former research assistant, who at the time, was conducting farming activities in one of the study areas. A formal study, part of which we publish in this paper, was inspired by that investigation. This qualitative paper combines community perspectives, and those of survivors of uvulitis after undergoing traditional uvulectomy, and of community health workers, who are the first level in the formal health system structure in Uganda [28]. It also includes perspectives of traditional surgeons, entrusted with the surgery in the communities. Such triangulated perspectives provide further guidance to programming aimed at supporting sufferers of uvulitis that need safe care. This paper therefore explores the beliefs and practices regarding uvula infection and traditional uvulectomy in Luwero district, central Uganda. The findings of this study can guide programming to improve response to this condition.

## Methods

### Study design, site and sample

This study forms part of a larger mixed method research [29] that investigated the burden of the condition called "*akamiro*" [uvula infection], its general epidemiology, beliefs and practices around it, and the traditional surgical procedure (uvulectomy). We draw on the qualitative description [30] for this paper to address the latter part of the objective; beliefs and practices regarding uvula infection and traditional uvulectomy. The study was conducted in Luwero district, central Uganda. The district covers an area of 2,218 square KMs with a population of 511,900 and population density of 230.8/ square KM. It comprises of 13 sub counties, 90 parishes and 594 villages. Two sub counties, Zirobwe and Bombo Town Council were selected for

this study, partly because the reported cases during an early investigation identified these as the epicentres of uvulectomy. The two sub counties are predominantly occupied by Baganda ethnic group, with Bombo Town Council including a large Nubian community.

The sample consisted of six focus group discussions (FGDs) with community members, 12 interviews with uvulectomy clients, six with traditional surgeons, and 10 key informant interviews (KIIs) with community health workers (CHWs). The CHWs in Uganda are known as Village health Team (VHT) members. They are incorporated into the formal health system as the lowest health delivery structure, also known as health centre 1. They comprise of volunteers that are selected by communities from within themselves. CHWs are offered on-job training and refresher sessions by Ministry of health and implementing partners in health. Their roles include supporting the primary healthcare function, offering health information, and linking community members to the health services. [28] The latter role would include services like referral of persons with Uvula infections, for example. In total, 14 were conducted in Bombo Town council and 20 in Zirobwe sub county.

## Sampling, data collection and tools

Participants were purposely selected from Zirobwe sub county and Bombo Town council, which were the sites for the larger survey. The participants were carefully selected to include persons who underwent traditional uvulectomy or caregivers (if children) for lived experiences, known traditional surgeons and CHWs who relate with community members, to establish treatment practices and referrals, and community members resident for more than 12 months to explore community beliefs. Persons who underwent traditional uvulectomy or the caregivers were identified from the household survey and asked if they wanted to share some more information about their experience. The CHWs are known in the villages, and were selected based on availability as long as they were active in their roles. Traditional surgeons were also known persons well identified by the community members. They were approached to voluntarily participate after explaining the study purpose. FGD members from the communities were selected with the support of the CHWs who supported mobilisation. They should have been residents of the community for at least 12 months. Data were collected in March 2021 by a team of four well trained and experienced research assistants (two females and two males). They were supervised by SPSK, and JN who also coordinated the study.

Interviews were mainly conducted in native Luganda language, and audio recorded with permission. Interview guides for CHWs included questions around their experiences dealing with households that reported cases of *akamiro*, approach to these cases including any advice provided and referrals, what they know about patterns of resort from their clients and recommendations on what to do with this condition. Traditional surgeons guide included topics around what they know about *akamiro*, including causes, symptoms and diagnosis, how they treat *akamiro*, views of their clients, patterns of resort they are aware of, their advice to the clients including any referrals, their take on the medical handling of *akamiro*, and details about the uvulectomy procedure. FGD guides included among others, what is known about *akamiro* including symptoms and causes, community perceptions about uvulectomy, how its diagnosed, community response when diagnosis is done, patterns of resort, uvulectomy procedure, who they trust to handle *akamiro*, uvulectomy costs, and any recommendations. Uvulectomy client interviews included their lived experience with *akamiro*, and the details from the uvulectomy procedure, the diagnosis, who they trust and any recommendations. Interviews with clients and surgeons were mainly to supplement the survey, with brief topic guides. These interviews with clients lasted an average of 30 minutes, while those with VHTs were shorter lasting an average of 25 minutes. FGDs lasted an average of 1 hour and 17 minutes.

### Data management and analysis

All interviews and group discussions were audio recorded using Sony digital voice recorders. Interviewers transcribed and translated their own recordings. SPSK and an independent research assistant conducted the coding. The two developed an analysis plan using the thematic areas based on the topic guides. We followed the thematic analysis steps [31] that included: (i) familiarizing ourselves with the data after transcribing by reading several transcripts; (ii) generating initial codes used to guide the rest of the codebook, which were included in the draft analysis plan that followed the topic guides and updated during the coding process; (iii) coding systematically across the entire data; (iv) collating data relevant to each code; and (v) iteratively identifying the themes by collating codes into potential themes and gathering all data relevant to each potential theme. The analysis included running query reports of the codes with their quotations, and a codes document table to ascertain frequency of responses. Coding and analysis were conducted using Atlas ti 9.

### Ethical considerations

The larger study was approved by the Makerere University School of Public Health, Research and Ethics Committee (Protocol 870), and by the Uganda National Council for Science and Technology (HS1167ES). All participants in the study provided written informed consent for interview and audio recordings. For the children who underwent traditional uvulectomy, their caregivers were interviewed. Permission was also sought from the district leadership before entry. Findings have been disseminated to the representatives of participants and the district health leadership, for action.

## Results

### Community understanding of uvulitis *(Akamiro)*

The study explored the understanding of Uvulitis that was referred to as "*Akamiro*", as well as the beliefs around its treatment. Community members described *Akamiro* using varied terms that mainly included: a big pimple, a large object the size of a chicken heart, one that grows larger compared to the normal and rests at the base of the tongue. Among the children, participants noted that it is an elongated uvula that is easily visible at the back of the mouth when a child is crying.

> *It looks like a chicken heart. It grows on the throat itself. . . elongates and develops pus. Every time you cough, it itches and if it bursts, the person dies. When it grows, it falls on the tongue. That is the thing they cut.* FGD 1_Zirobwe

> *It is a pimple. It is longer than these pimples that attack the skin. That is why when they cut it off, it comes with a piece of meat.* FGD1_Bombo

All participants including traditional surgeons acknowledged that they did not know what causes this condition. They believed that it was naturally occurring, even among babies. They reported its symptoms to include persistent and discomforting cough that eventually leads to loss of appetite and weight loss. Particularly, the children with this condition persistently struggle to swallow food or milk. Breastfeeding infants vomit the breast milk, which is believed to be caused by chocking from the elongated uvula. In some cases, the vomiting was also believed to result from a fluid that the elongated uvula emits when the child coughs. Lived experiences from in-depth interviews with survivors indicated developing pass, and swelling of the stomach that they believed resulted from swallowing the fluid generated by the elongated infected

uvula. The uvula was also linked to an overflow of saliva, fever, difficulty in breathing and speech, diarrhoea among children, and vomiting.

> I would cough and vomit, at times I would get itching in the neck and could not eat. I would also get a lot of neck and chest pain. It lasted for about two weeks. Even when I opened my mouth and the air enters I would feel bad. It was in the second week that I got a traditional surgeon. I went to him and he operated my uvula. When he finished he gave me salt water mixture to drink and some herbal medicine, and I went back home. When I came back I took food well without the pain I went through before the uvulectomy. The pain reduced after four days and I got my peace back. From that time, I have never felt any pain in the neck again or chest, and coughing stopped. I am fine. Uvulectomy client 1_Zirobwe

> . . .It is dangerous because you do not get peace when you have it, it causes continuous severe coughing, one coughs nonstop without stopping, you have no peace until it is cut out and one gets better, at night one can't sleep. Traditional Surgeon 1_ Bombo

## Community diagnosis of *Akamiro* and decision to seek uvulectomy

The diagnosis of *Akamiro* was initially made by the caregivers or the patient experiencing the above symptoms, especially persistent cough, fever, and difficulty in swallowing. S/he then consults the trusted persons in the community to confirm. These included the known traditional surgeons followed by one of the community elders, known as *omukebezi*, and then the formal system health workers. The significant others like elderly grandparents and some selected neighbours were involved sometimes.

> . . .the parents know it, the moment you cough for some days, they tell you, open your mouth and we see, because you can see it. They just use a torch and they see that it is the one. Community health worker 3_Bombo

> One old woman [omukebezi] in the village examined her and told me that she had uvula that is why I took her to the local surgeon. She is specialized in identifying 'Akamiro'. Me I could not tell that the child had 'Akamiro'. Uvulectomy client /caregiver 5_Zirobwe

When it is confirmed that the patient has *akamiro*, most participants agreed that the traditional surgeons were the most trusted people to provide solutions. Surgical removal was the best option. *Akamiro* was perceived as fatal if not removed in time.

> If surgery is not done, the child does not eat or swallow. I saw a 2 months old child who was suffering from it. They took him to a traditional surgeon who performed surgery. . . The child cannot swallow. It may even choke them. FGD 3_Bombo

Traditional surgeons also in line with other participants, confirmed that the only way to get relief from *akamiro* was to undergo surgery. They also concurred with most community participants that the formal health facilities in the area struggled to manage this condition, leaving people with mainly the traditional surgeons as to help them.

> Doctors at the hospital have not reached the level of treating it. They claim that they don't know the disease at all. Traditional Surgeon 3_Bombo

> I have never heard that when you use such and such [western] medicine you can get better. No. But when they cut it that is when you get peace. Traditional Surgeon 1_Bombo

The reported pattern of resort for *Akamiro* was hierarchical in majority cases. The patients first reported to public health centres and a few to private clinics before being taken to traditional surgeons. This pattern of resort was dictated by the main symptom, which is persistent cough that often requires the formal health system. It was reported that the patients are treated for chronic cough and after persistent symptoms, they are taken to traditional surgeons. From varied accounts of both community members and CHWs, they pointed to a limitation with the formal health system in the area where the health workers *"did not understand"* the condition. Thus, the hierarchical pattern of resort is influenced by the failure to receive adequate satisfactory care at the health centres. Many participants noted that the best care and lasting treatment for Akamiro comes from the traditional surgeons. The response was also influence by the advice from those who have had or heard about the condition. They advised about the alternative when the formal health system failed. Most CHWs also noted that although they refer clients to the formal health system, they end up at the traditional surgeons after. Others noted that they are approached to treat cough that persists, forcing clients to seek further advice.

> *My child has suffered from it. I went to the hospital to seek treatment, but the problem is that the trained health workers have no knowledge about it. They present with fever and excessive cough, until you get a person who advises you to go to a "mukebezi" to check if it [Akamiro] is there. When she checks and it is actually there, she refers you to a traditional surgeon in* Luwero *[concealed location name]. . . He is famous for uvulectomy. After surgery, the fever reduces, although the child cannot breast feed well immediately.* FGD 2 Zirobwe

> *They do not bring the child and say it is uvula. We usually handle children under 5 years, they bring the child with cough, so I give the child medication for cough. I usually ask for the age of the child and for how long the child has coughed, then I give the medication for cough which is amoxicillin. It is after failing to heal that they decide to take the child to the [traditional] surgeon. So, to me they bring somebody suffering from cough, not uvula.* Community Health Worker 2, Zirobwe

## The traditional uvulectomy procedure

The community reported varied accounts regarding the traditional surgical procedure. It is either carried out in the early morning hours or after sunset. Most traditional surgeons were reported to conduct the surgery in the morning hours.

> *I went at 4:00pm. Each surgeon has different timing. Now, for the case where I took my first child, he operates in the morning before the sun heats up. When the sun is shining, they do not operate since at the time the blood is flowing fast. Most of them operate in morning hours because the blood flow is not so fast.* Uvulectomy client 7, Zirobwe

> *I informed him [surgeon] in the morning, and he told me that they usually cut this thing during morning hours or in the evening when moon is up.* Uvulectomy client 4, Bombo

## Duration of the uvulectomy procedure

We explored how long the procedure takes and the time spent at the traditional surgeons' place. Most participants reported that they or other people they knew spent a short time at the surgeons, usually less than an hour. The client heads home soon after the uvulectomy

procedure. The community members also noted that the actual procedure usually takes a minute or less with experienced traditional surgeons, with surgeons also confirming this timing.

*. . .To me, if you have not disturbed me, within a minute I may have finished you. Babies don't disturb because if a mother holds it properly the way you have instructed her. . . within one minute I am done.* Traditional Surgeon 1_Bombo

## Traditional uvulectomy tools

The main tools reported by community members, CHWs and the surgeons themselves were new razor blades, sticks, reeds, a piece of wood or a small board, strings, wires, curved sharp knives, cotton wool, and spoons. In a few cases, gloves were reported by some traditional surgeons. The razor blade was the most used item, broken into two to four pieces, intentionally small enough to enter the mouth of the patient without causing unintended injury. The razorblade is used together with either a reed or another type of stick, and a string or wire to hold the uvula. In some cases, a spoon, a piece of wood or a small board were also used in the process to help the patient keep an open mouth, hold the tongue down, and later hold the uvula. Aside from a razor blade that some clients reported went with, the rest of the tools were found at the traditional surgeons. A few surgeons also reported using a small curved sharp knife in lieu of a razor blade.

The quotations illustrate the typical explanations from participants:

*The razor blade was new, he broke it into four small pieces that can fit in the mouth of the child. Then he put that small piece on a stick so that he could be able to hook the uvula and cut it. Then he put a spoon in the mouth of the child to keep it open.* Uvulectomy client 7, Zirobwe

*I get a piece of wood which helps to press and hold the tongue down, and then the Uvula is clearly seen. All of it rests on the wood. So, when you cut, it just comes out [with the wood]. I use a curved knife to cut it, pressing it against the wood, I squeeze the knife to cut it down. . . . It is a small knife just similar to a nail.* Traditional Surgeon 1_Bombo

*He brought a stick; it was like a reed. He broke a razorblade and attached it to the reed. . . .. I cannot explain it because I had just opened my mouth. But when he was finishing, I saw he had broken the razorblade. There is a way he cut [cannot explain it well], he told me to widely open my mouth. So, when I opened, he cut.* Uvulectomy client 2_Zirobwe

*He says that if he cuts the uvula and you swallow it, you die. So, he uses a string to tie it, the moment he cuts it, he pulls it out. He first hooks it before cutting it.* Uvulectomy client 4_Zirobwe

Traditional surgeons also noted that they used gloves during the surgery, primarily to protect themselves from infections. However, only one client supported the use of gloves claimed by surgeons.

*I use gloves. It is me who protects myself from getting infected because we are local surgeons in the community. You cannot know whether the patient has diseases like TB, you cannot know. We have to do it very fast. When a patient opens the mouth, you just cut it out very fast such that the patient goes away quickly without infecting you with his/her disease.* Traditional Surgeon 1_Bombo

After the procedure, items like cotton wool, salty foods like corn were advised. The cotton wool was inserted in the ears that participants believed was to *'prevent air entering and affecting the surgical site'*. The surgeons explained to the clients that the ear, nose throat is linked.

P2: the cotton prevents cold air from going into the wound after surgery.

P3: *Once, the cold air enters and blows over the fresh wound, it pains a lot. You can close the mouth to prevent air from entering, but you cannot close the ears. That is why he inserts a piece of cotton wool*. FGD 1_Zirobwe

## The cost of traditional uvulectomy

There was no standard cost reported for the traditional surgery. Traditional surgeons charged variably basing on perceived social economic status of the client. The charges ranged from 3,000 to 100,000 Uganda shillings [about 90 US cents to 28 US dollars]. In case clients had no cash, they reported to sometimes pay in kind with items like chicken, goats and beer, all at will.

*They are highly experienced surgeons. When it comes to the cost, the children are charged 30,000/ = to 50,000/ =, adults are charged 70,000/ =. It depends on the financial situation you are in that time*. FGD 2, Zirobwe

*Sometimes they just give us chicken, and you also take it. Your aim is to heal a person*. Traditional Surgeon 1, Zirobwe

Some surgeons reported caretakers of children that needed urgent surgery but did not have money were not charged. They reported to sometimes offer free services to save the life, with some people coming back later to express gratitude with whatever they could afford.

*There are those who come when they do not have the money, you help them out. One may come with a child and the condition of the child is not good at all, then you just have to be kind hearted and help them because you are also a parent*. Traditional Surgeon 1_Bombo

## Community trust in traditional surgeons

The community members expressed near complete trust in traditional surgeons, who they perceive as the *'only persons that can perform uvulectomy well'*, and have been in the community for a long time. Participants said they had not heard of cases of death directly at the hands of surgeons reported, which further entrenched their position. Surgeons were seen as having a special God given talent, while other people also equated them to traditional circumcisers that have built a reputation from the experience over time. The cost of procedure and the flexibility in payment including in kind and in rare cases offering free services to save lives were further justifications for their utmost trust. This, coupled with a speedy procedure and predictable good outcomes ensured they gained more trust. They were also readily available when needed.

Since the traditional surgeon is the one who knows it, then you can trust them to treat it. Just like the health workers who treat Malaria, they are also fully trusted. FGD 2_Bombo

*We trust him because he has treated very many people and none of them has died. People go to hospitals for surgery, patients die after but with him, nobody has died*. FGD 1_Zirobwe

Community members intimated that the experience from some formal health facilities where people resorted to first for care was not satisfying. This also entrenched the position of the traditional surgeons who were able to cure the failed cases from the formal health system. The community members also reported that their traditional surgeons had several cases brought to them from outside the community. With such demand, their trust in the community was further strengthened.

*It is not easy to treat. Not everyone one can do it. . . In this area, we have got only one traditional surgeon. People come from all over. If there was as traditional surgeon in Kikyusa and Kyaggwe, those people there would not be coming this side.* FGD 1_Zirobwe.

Despite the overwhelming trust by most participants, there were deviant cases of scepticism about the traditional surgeons. However, these were rare, contextual and not completely negative. A few people said they could not trust those who are new, while one person who underwent a successful surgery said they would not advise someone to go there now, because times have changed. There were fears of potential infections resulting from non-use of gloves and bad tools. Worth noting is that one surgeon reported also reported that there are colleagues he does not trust.

*Others are there but I don't trust them, they try but. . . no they are fake. They try, sometimes they cut and cause more harm. Sometimes a person is hurt and abandoned, so he looks for another surgeon to help treat him.* Traditional Surgeon, 2_Bombo

The CHWs also trusted the traditional surgeons, and most confessed to referring clients to both the formal health centres but also to traditional surgeons; even their own relatives. Those who had not referred them yet, said they would refer them in future, if approached. CHWs noted that it was better for clients to go the traditional surgeons where they get assured quick services than the formal health system where several referrals will be made from facility to another until the National referral hospital level. Some believed that there is no other formal healthcare solution to the *akamiro* problem.

*. . .because even when one comes here [health centre], he will make a line and they will end up referring him there [another facility], even there, they will write for him a letter referring him to Mulago [National referral hospital]. So, when he sees that process of being disturbed, he just cuts it short and goes there [traditional surgeon]. The people they treat, get healed. For example, my brother, he was cut and now he is healed.* Community health worker 1_Bombo

*I would just direct them to go for uvulectomy since there is not any other treatment for it. Yes, I know the [traditional] surgeon, so if a patient came to me, I would direct him to the surgeon.* Community health worker 1_ Zirobwe

## Discussion

This study indicates that Uvula infection, known as *Akamiro* and the associated traditional uvulectomy are common in Luwero district. *Akamiro* was referred to as a big pimple, an object the size of a chicken heart, a swelling visible when a child cries, or a larger than normal uvula, whose causes were unknown. However, symptoms were clearly known including persistent cough, diarrhoea, vomiting, loss of appetite, inability to swallow, chocking on breast milk or food, ultimately weight loss, swollen stomach, overflow of saliva, fever, difficulty in breathing

and speech. *Akamiro* diagnosis was confirmed after seeking care from health workers or in consultation with significant others like grandparents, community elder and finally the traditional surgeon. The main pattern of resort was hierarchical, from health centres to traditional surgeons. The surgery lasted only a short period, conducted before the heat of the day; in the morning or after sun set, using a variety of tools like razor blades, reeds, strings, wires, curved special knives and spoons. Traditional surgeons offered flexibility in costing clients and was either cash or in kind, and had immense trust from the community including from the CHWs.

The causes of *akamiro* were unknown. None of the participants including the traditional surgeons in this study indicated knowledge about causes, even in general terms. From the literature, uvulitis has been attributed to infectious and non-infectious causes [4–9]. However, participants in the current study were able to diagnose Uvula infection using symptoms such as persistent cough, inability to swallow food of breast milk, overflow of saliva among others. Some of these are similar to symptoms reported in other countries such as Ethiopia [32], Tanzania [21], Nigeria [25] and Niger [18]. This implies shared beliefs across cultures in multiple countries on the continent.

There are several tools used in conducting uvulectomy [25, 32]. None of the surgeons in our study had any western equipment, except a razor blade. They all used traditional tools, some of which have also been reported several decades ago [33]. In a Tanzanian study, they had tools similar to those in the formal health facilities including pairs of forceps [21], but we did not find any such. However, the time it takes to perform the surgery was almost similar to that reported in the Tanzania study, which was about 3–4 seconds.

We found a hierarchical pattern of resort where caregivers of children or adults' clients sought care from the professional sector, before finally going to traditional surgeons for uvulectomy. This reinforces the other finding that formal health workers at lower levels in the study area did not know how to handle this condition, as reported by both community members and traditional surgeons. Heading further to traditional surgeons was because of limited knowledge and thus support from the formal health system, yet people preferred to go there first. This also concurs with other studies [21, 32] that reported the hierarchical pattern of resort as well as limited knowledge from the professional sector. Participants in our study reporting that *akamiro* to be first treated in the health centres before heading further on, shows that the public has trust in the professional sector. Improvements in managing this condition in the professional sector, including proper referrals to the levels where clients can be helped will go a long way in improving safety for clients.

While in some settings, it has been reported as a clandestine practise [34], in this study, the practice is overtly discussed and traditional surgeons are well known by the community. CHWs interviewed referred clients to traditional surgeons when they noticed the symptoms indicated in this study. Such referrals by the village level health workers provides trust in the traditional surgeons. The community trusted uvulectomy with the folk practitioners. These have no formal training to conduct the practice but have won the trust of the community over time with no major negative incidences reported by the community. In a recent study in Tanzania [21], the practitioners reported several ways of winning trust of the community members, including handling them with care, and not being greedy. In this current study the surgeons reported to be flexible in charging clients, both cash and kind, including conducting uvulectomy freely for needy clients. Patient or caregiver trust in informal sector practitioners in Uganda has been reported before for other conditions [35].

Good outcomes of the procedure [32] with no complications reported could also cement the place of traditional surgeons as the go to, for uvulectomy. In a dissemination meeting we held in Luwero town with the district leadership, traditional surgeons, VHTs, and local leaders from the two sub counties, the surgeons confirmed that they have not had any complications.

Similar sentiments have been reported among traditional practitioners [21] and caregivers [32], although earlier studies indicated there are several complications from traditional uvulectomy [20, 24–27, 36] including mortality [24]. In a study in Nigeria, 58% of patients did not clear symptoms that led to uvulectomy [25].

The strength of this study is in triangulation [37] of methods and sources. This enhances the rigour of these findings. Reporting vast similarities in responses from the community members, traditional surgeons and community health workers provides credibility to the findings. The limitation could be from reports of traditional surgeons and community health workers who may provide desirable answers based on their work. Community health workers may under report referrals to the traditional surgeons because they are expected to refer within the formal health system. Traditional surgeons may also underreport complications arising from uvulectomy or even fear to expose their tools to researchers. However, the triangulation of data would have exposed such biases. We also introduced the study openly to all groups of participants and written informed consent was obtained with clear explanation of the purpose of the study. There is a risk of loss of information during the transcription as translations are done from Luganda to English, given that the transcribers are not native English speakers. A quality control was done by SPSK, although this may not entirely eliminate the risk.

This study provides scientific evidence and forms a basis for further investigations on uvula infections and traditional uvulectomy. The failure of the health centres to help persons reporting uvula infections drove people into traditional uvulectomy, whose cost was relatively affordable and the care from traditional surgeons was well perceived. The CHWs enhanced the role of traditional surgeons with referring clients to them, after noticing limited support in the formal health system. However, the tools used are non-sterilised equipment and local objects that can cause harmful. The Ministry of Health through its district health team structures can devise ways to handle the uvula infections that lead to traditional uvulectomy. One way is through community health sensitization, encouraging populations to seek formal medical care in public health centres. However, with evidence of limited to no information about uvulitis from by the lower level health workers, this cannot be a success if their training and sensitization is not conducted.

## Supporting information

**S1 Text. Excerpts from focus group discussions with community members.**
(DOCX)

**S2 Text. Excerpts from key informant interviews with community health workers.**
(DOCX)

**S3 Text. Excerpts from transcripts of IDIs with uvulectomy clients and caregivers.**
(DOCX)

**S4 Text. Excerpts from transcripts of IDIs with traditional surgeons.**
(DOCX)

## Acknowledgments

We appreciate the following persons: Stellah Kayongo, who alerted the Makerere University School of Public Health team about the existence of traditional uvulectomy in the community, for her vigilance; the research assistants who collected the data; the community members, CHWs and traditional surgeons for their willingness to share their experiences; and the district leadership in Luwero for their cooperation.

## Author Contributions

**Conceptualization:** Simon Peter Sebina Kibira, Juliana Namutundu, Noah Kiwanuka, Victoria Nankabirwa, Justine Namwagala.

**Data curation:** Simon Peter Sebina Kibira.

**Formal analysis:** Simon Peter Sebina Kibira.

**Funding acquisition:** Juliana Namutundu, Noah Kiwanuka.

**Investigation:** Simon Peter Sebina Kibira, Juliana Namutundu, Noah Kiwanuka, Victoria Nankabirwa, Justine Namwagala.

**Methodology:** Simon Peter Sebina Kibira, Juliana Namutundu, Julius Kiwanuka, Noah Kiwanuka.

**Project administration:** Juliana Namutundu, Noah Kiwanuka.

**Software:** Simon Peter Sebina Kibira.

**Supervision:** Simon Peter Sebina Kibira, Juliana Namutundu.

**Validation:** Juliana Namutundu, Julius Kiwanuka, Victoria Nankabirwa.

**Writing – original draft:** Simon Peter Sebina Kibira.

**Writing – review & editing:** Juliana Namutundu, Julius Kiwanuka, Noah Kiwanuka, Victoria Nankabirwa, Justine Namwagala.

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
