## [Decision Letter · Decision Letter 0]

18 Apr 2023

PGPH-D-23-00282

Uvula Infections and Traditional Uvulectomy: Beliefs and Practices in Luwero District, Central Uganda

Dear Dr. Kibira,

Thank you for submitting your manuscript to PLOS Global Public Health. After careful consideration, we feel that it has merit but does not fully meet PLOS Global Public Health’s publication criteria as it currently stands. Therefore, we invite you to submit a revised version of the manuscript that addresses the points raised during the review process.

Please note that we have only been able to secure a single reviewer to assess your manuscript. We are issuing a decision on your manuscript at this point to prevent further delays in the evaluation of your manuscript. Please be aware that the editor who handles your revised manuscript might find it necessary to invite additional reviewers to assess this work once the revised manuscript is submitted. However, we will aim to proceed on the basis of this single review if possible. 

We look forward to receiving your revised manuscript.

Kind regards,

Steve Zimmerman, PhD

PLOS Staff Editor

Journal Requirements:

Additional Editor Comments (if provided):

Reviewers' comments:

Reviewer's Responses to Questions

**Comments to the Author**

1. Does this manuscript meet PLOS Global Public Health’s publication criteria? Is the manuscript technically sound, and do the data support the conclusions? The manuscript must describe methodologically and ethically rigorous research with conclusions that are appropriately drawn based on the data presented.

Reviewer #1: Partly

2. Has the statistical analysis been performed appropriately and rigorously?

Reviewer #1: N/A

3. Have the authors made all data underlying the findings in their manuscript fully available (please refer to the Data Availability Statement at the start of the manuscript PDF file)?

Reviewer #1: Yes

4. Is the manuscript presented in an intelligible fashion and written in standard English?

Reviewer #1: Yes

5. Review Comments to the Author

Reviewer #1: This article draft looks at practices related to traditional uvulectomy, a phenomenon that appears to be relatively common in several African countries, but that is under-researched. The authors write that to the best of their knowledge, this is the first study to look at this form of folk surgery in Uganda. The article is therefore an important contribution to this field of study.

The article does not relate to a conceptual framework or theory. The main strength of the article draft is the triangulation, which gives us the perceptions of both the folk surgeons, community members, uvulectomy clients, as well as community health workers. The findings are very interesting, and they are discussed in relation to existing literature on traditional uvulectomy.

I highly recommend this article draft for publication, but some issues need to be addressed before publication:

Ethics: In the transcripts of the interviews, the full personal names of all the participants are given. Did they give their consent to this? If yes, have the researchers considered any negative consequences, particularly for the lay surgeons? Even if the practice is not restricted by law today, that may happen later (not the least after the publication of this article). I recommend that all the personal names are removed from the transcripts if they are made publicly available.

Community health workers: Please provide some more background information about this cadre in the Ugandan health system since the term “community health worker” is used for both salaried and volunteer health workers in different contexts. What kind of training/education do they have? Are they based at local level health facilities or do they just operate from their homes?

Methods:

Page 5, line 101-15: The authors state that the two sub-counties are “predominately occupied by Baganda” (…) “with Bombo Town Council including a large Nubian community”. Did the authors ask the participants about their ethnic affiliation? It would be useful to have some reflections regarding the degree to which the uvulectomy practices are linked to certain ethnic groups or not.

Page 5, line 106-109. The authors describe all the interviews as “in-depth”, but do not say anything about the length of the interviews. Information about the length should be included in the revised manuscript (no need to state the exact length of each interview and FDG, but a time range should be given).

From the attached transcripts, we see that of the eight transcribed interviews with community health workers, four of them are only 3-4 pages long. Similarly, of the IDIs with clients, three of the eight interviews are 5-6 pages long. Normally, to qualify as an in-depth interview, the interview should last approximately an hour (a minimum of 30 minutes), and if this had been the case the transcript would be much longer in my experience.

I also note that the FDGs in Zirobwe are considerably longer than those in Bombo, and in Bombo, the participants generally give very brief responses. This makes me wonder whether this is linked to the fact that different research assistants worked in the two locations. I also wonder whether the transcripts are indeed verbatim (word by word) or whether the research assistants perhaps summarised the interviews in the original language (Baganda) during the translation process to English? Did any of the authors listen to the recordings to check the quality of the translated transcriptions? Perhaps some reflections on this can be added to the limitations section.

Page 5, line 112-113: “The participants were carefully selected to include…”. Exactly how were the participants recruited?

Page 6, line 120-134: This section describes the issues that were covered in the topic guides but does not say anything about why the researchers decided to ask these specific questions. For example, from the transcripts we see that participants were asked about the possible connection between TB and akamiro. Was this based on a pre-study, information from Stellah Kayongo (“who alerted the university about the existence of traditional uvulectomy”, page 21), or was it based on reading of the existing literature on traditional uvulectomy?

Language: Overall, the manuscript is very well written and structured. However, there are a few errors that need to be corrected (please note that there may be others, since this reviewer is not a native English speaker).

Page 2, line 36: “using audio recorder key informants interviews” – should be “recorded”?

Page 2, line 41: “the sized chicken heart” – should be “the size of a chicken heart”?

Page 5, line 117: “by a team four well…” – should be “team of four…”?

Page 11, line 236: “My child has ever suffered from it” – should be “has suffered from it”?

Page 13, line 295: “there is away he cut” . should be “a way”?

I wish the authors all the best for these minor revisions and look froward to seeing the published version.

6. PLOS authors have the option to publish the peer review history of their article (what does this mean?). If published, this will include your full peer review and any attached files.

**Do you want your identity to be public for this peer review?** For information about this choice, including consent withdrawal, please see our Privacy Policy.

Reviewer #1: No

---

## [Decision Letter · Decision Letter 1]

26 May 2023

Uvula Infections and Traditional Uvulectomy: Beliefs and Practices in Luwero District, Central Uganda

PGPH-D-23-00282R1

Dear Dr. Kibira,

We are pleased to inform you that your manuscript 'Uvula Infections and Traditional Uvulectomy: Beliefs and Practices in Luwero District, Central Uganda' has been provisionally accepted for publication in PLOS Global Public Health.

Best regards,

Susan Julia Chand, PhD

Guest Editor

Reviewer Comments (if any, and for reference):

Reviewer's Responses to Questions

**Comments to the Author**

1. If the authors have adequately addressed your comments raised in a previous round of review and you feel that this manuscript is now acceptable for publication, you may indicate that here to bypass the “Comments to the Author” section, enter your conflict of interest statement in the “Confidential to Editor” section, and submit your "Accept" recommendation.

Reviewer #1: All comments have been addressed

2. Does this manuscript meet PLOS Global Public Health’s publication criteria? Is the manuscript technically sound, and do the data support the conclusions? The manuscript must describe methodologically and ethically rigorous research with conclusions that are appropriately drawn based on the data presented.

Reviewer #1: Yes

3. Has the statistical analysis been performed appropriately and rigorously?

Reviewer #1: N/A

4. Have the authors made all data underlying the findings in their manuscript fully available (please refer to the Data Availability Statement at the start of the manuscript PDF file)?

Reviewer #1: Yes

5. Is the manuscript presented in an intelligible fashion and written in standard English?

Reviewer #1: Yes

6. Review Comments to the Author

Reviewer #1: The authors have addressed all my comments in a satisfactory manner.

7. PLOS authors have the option to publish the peer review history of their article (what does this mean?). If published, this will include your full peer review and any attached files.

**Do you want your identity to be public for this peer review?** For information about this choice, including consent withdrawal, please see our Privacy Policy.

Reviewer #1: **Yes: **Siri Lange
